# A Novel P53 Nanomedicine Reduces Immunosuppression and Augments Anti-PD-1 Therapy for Non-Small Cell Lung Cancer in Syngeneic Mouse Models

**DOI:** 10.3390/cells11213434

**Published:** 2022-10-31

**Authors:** Sang-Soo Kim, Joe B. Harford, Manish Moghe, Caroline Doherty, Esther H. Chang

**Affiliations:** 1Department of Oncology, Lombardi Comprehensive Cancer Center, Georgetown University Medical Center, Washington, DC 20057, USA; 2SynerGene Therapeutics, Inc., Potomac, MD 20854, USA; 3College of Medicine and Science, Mayo Clinic, Rochester, MN 55905, USA

**Keywords:** anti-PD-1, immune checkpoint inhibitor, immunosuppression, non-small cell lung cancer, nano-medicine, p53, SGT-53, tumor-targeted delivery

## Abstract

Lung cancer is among the most common and lethal cancers and warrants novel therapeutic approaches to improving patient outcomes. Although immune checkpoint inhibitors (ICIs) have demonstrated substantial clinical benefits, most patients remain unresponsive to currently approved ICIs or develop resistance after initial response. Many ongoing clinical studies are investigating combination therapies to address the limited efficacy of ICIs. Here, we have assessed whether *p53* gene therapy via a tumor-targeting nanomedicine (termed SGT-53) can augment anti-programmed cell death-1 (PD-1) immunotherapy to expand its use in non-responding patients. Using syngeneic mouse models of lung cancers that are resistant to anti-PD-1, we demonstrate that restoration of normal p53 function potentiates anti-PD-1 to inhibit tumor growth and prolong survival of tumor-bearing animals. Our data indicate that SGT-53 can restore effective immune responses against lung cancer cells by reducing immuno-suppressive cells (M2 macrophages and regulatory T cells) and by downregulating immunosuppressive molecules (e.g., galectin-1, a negative regulator of T cell activation and survival) while increasing activity of cytotoxic T cells. These results suggest that combining SGT-53 with anti-PD-1 immunotherapy could increase the fraction of lung cancer patients that responds to anti-PD-1 therapy and support evaluation of this combination particularly in patients with ICI-resistant lung cancers.

## 1. Introduction

Lung cancer is one of the most common malignancies and the leading cause of cancer-related death in the world [1]. Lung cancer is usually diagnosed at late stages, resulting in poor survival rate [1]. For many decades, platinum-based combination chemotherapy was the only systemic treatment available for advanced non-small cell lung cancer (NSCLC), and this treatment regimen yielded only a modest increase in survival accompanied by very high toxicity [2]. The discovery of oncogenic driver mutations (e.g., epidermal growth factor receptor (EGFR)) was the first major breakthrough that led to the development of molecularly targeted therapies. However, EGFR inhibitors improved survival for only the small subset of lung cancer patients having tumors that harbor EGFR mutations [2]. Moreover, drug resistance inevitably developed in these patients, and nearly all who initially responded to EGFR inhibitors developed progressive disease within approximately one year of starting treatment [3]. There are clinical limitations of currently recommended therapies for patients with advanced NSCLC, and the quest for novel therapeutic approaches to improve patient outcomes remains a priority.

Another major breakthrough in treating cancers generally has been the emergence of immune checkpoint inhibitors (ICIs) including antibodies targeting cytotoxic T-lymphocyte-associated protein 4 (CTLA-4) or programmed cell death-1/programmed cell death ligand 1 (PD-1/PD-L1). ICIs produce robust and durable responses in patients with advanced solid tumors and have transformed the cancer care paradigm [4]. However, ICIs as monotherapies have met with only limited success and a large number of lung cancer patients either present with or develop therapeutic resistance [5]. Resistance to ICIs is apparently due, in part, to the fact that tumors are able to modify the tumor microenvironment (TME) to suppress and evade immune responses [6]. Thus, strategies aimed at restoring antitumor immune responses have the potential to improve ICI therapies to benefit more patients with lung cancer.

Currently, a large number of trials combining ICIs with various other therapeutic modalities (i.e., chemotherapeutics, radiation, and molecularly targeted therapies) are underway that all aim to enhance treatment efficacy of ICIs [7]. The approach that we are taking in this regard combines ICI with gene therapy that restores the functions of the tumor suppressor p53 via SGT-53, an investigational tumor-targeting nanomedicine comprising a novel cationic immunoliposome encapsulating a plasmid DNA carrying the wild-type *TP53* gene. When systemically administered, SGT-53 delivers its plasmid payload to tumor cells with a very high specificity and efficiency via a single-chain antibody fragment (scFv) recognizing transferrin receptors (TfRs), which are overexpressed on the surface of virtually all cancer cells [8]. The p53 protein encoded by *TP53* is a pleiotropic regulator of diverse cellular pathways, and *TP53* gene is frequently altered in various types of cancer. For the lung cancers, a dysfunctional p53 pathway is particularly prevalent with ~68% of NSCLC patients having an altered p53 pathway [9]. While p53 has been most extensively studied as a tumor suppressor that regulates the cell cycle and drives apoptosis, its role in modulating antitumor immune response has also been documented [10,11]. For example, a recent study demonstrated that tumor-specific loss of p53 leads to a modulation of myeloid and T cell responses and delays tumor rejection in immune-competent hosts [11]. On the basis of p53’s known participation in immune responses, we hypothesized that restoring p53 functions would augment antitumor immunity and enhance the efficacy of ICIs in treating NSCLCs, which are all too often refractory to such therapy.

The goal of our study is to assess whether restoring p53 function via the tumor-targeting nanomedicine SGT-53 would convert otherwise unresponsive lung cancers into tumors that are less refractory to anti-PD-1 immunotherapy. Here, we show that SGT-53 can restore effective immune responses against lung cancer cells by reducing immunosuppression through *galectin-1* (*Gal-1*) pathway previously shown to contribute to lung cancer progression [12]. Combining SGT-53 with anti-PD-1 therapy resulted in a significant augmentation of antitumor efficacy with a marked survival benefit in syngeneic mice bearing Lewis lung carcinoma (LL/2). These findings suggest that SGT-53 could boost antitumor immunity and thereby improve anti-PD-1 therapy for NSCLC patients.

## 2. Materials and Methods

### 2.1. Cells

LL/2 (RRID: CVCL_4358) and H358 (RRID: CVCL_1559) were obtained from ATCC (Manassas, VA, USA). Luciferase-expressing LL/2-luc cells were kindly provided by Dr. Raphael Nemenoff (University of Colorado). Cells were maintained at 37 °C in 5% CO_2_ in DMEM (LL/2 and LL/2-luc, Mediatech, Manassas, VA, USA) or RPMI (H358, Mediatech, Manassas, VA, USA) supplemented with 10% fetal bovine serum (Sigma, St. Louis, MO, USA). All experiments were performed with mycoplasma-free cells.

### 2.2. Nanocomplex Preparation

Cationic liposomes, referred to as Lip, consisting of 1,2-dioleoyl-3-trimethylammonium propane and dioleolylphosphatidyl ethanolamine (Avanti Polar Lipids, Alabaster, AL, USA) were prepared as described previously [13]. Plasmid DNA carrying the human wild-type *TP53* gene was encapsulated in TfRscFv/Lip (termed scL for single chain Liposome) nanocomplex (scL-p53, also known as SGT-53) [13]. For animal injections, 5% dextrose was added to nanocomplex preparation.

### 2.3. Transfection

LL/2 and H358 cells were plated at 6.0 × 10^5^ cells/10 cm dish for 24 h before transfection. SGT-53 nanocomplexes were diluted in serum-free media and added to dishes (7 µg of DNA/dish). After incubation for 5 h at 37 °C, the medium was replaced with 10 mL of fresh complete medium and the cells were further incubated.

### 2.4. Animal Studies

All animal experiments were performed in accordance with Georgetown University GUACUC protocols. For the tumor model with subcutaneously established tumors, C57BL/6J mice (6-week-old, female, Jackson Laboratories, Bar Harbor, ME, USA) were injected on their flanks with LL/2 cells (0.5 × 10^6^ cells/mouse). For the tumor model with lung metastatic tumors, C57BL/6J mice were injected with LL/2-luc cells (0.5 × 10^6^ cells/mouse) via intravenous (IV) injection into the lateral tail veins. Mice with established tumors were treated twice weekly with either SGT-53 (30 µg DNA/mouse, IV), with anti-PD-1 antibody [RMP1–14, BioXCell, Lebanon, NH; 200 µg/mouse, intraperitoneal (IP) injection] or with the combination of both of these agents. In the subcutaneous tumor model, at the time of harvest on day 25, mice were euthanized by CO_2_ asphyxiation and the peripheral blood and tumor were harvested. The peripheral blood was collected into a heparin-coated tube (BD Vacutainer) by cardiac puncture and centrifuged at 2000 rpm at 4 °C for 10 min to isolate plasma. Plasma samples were stored at −80 °C until analysis. The tumor was split with half immediately used to flow cytometry analysis and another half cut into two fragments for immunohistochemistry which was fixed in formalin and RNA analysis which was stored at −80 °C until extraction.

### 2.5. ATP Assay

Level of extracellular ATP in the cell culture medium was measured by luciferin-based ENLITEN ATP Assay (Promega, Madison, WI, USA) following the manufacturer’s instructions.

### 2.6. HMGB1 Assay

Concentrations of the high mobility group box 1 (HMGB1) protein in the cell culture media were measured following the manufacturer’s instructions using the HMGB1 enzyme-linked immunosorbent assay (ELISA) kit (Biomatik, Wilmington, DE, USA).

### 2.7. Gal-1 Immunoassay

Concentrations of the Gal-1 protein in the collected plasma or cell pellet samples were assayed using mouse Gal-1 ELISA kit (R&D Systems, Minneapolis, MN, USA) following the manufacturer’s protocols.

### 2.8. Immunohistochemistry

Tumors were fixed in formalin and embedded in paraffin block for immunohistochemistry (IHC). Tumor sections were stained using antibodies for Ki-67 (Dako, Denmark), cleaved caspase-3 (Casp3, Cell Signaling Technology, Danvers, MA, USA), CD8 (ThermoFisher, Waltham, MA, USA) or Gal-1 (R&D Systems, Minneapolis, MN, USA). Images were captured using Olympus DP70 camera on Olympus BX61 microscope. Captured images were analyzed using IHC Profiler plugin in ImageJ.

### 2.9. Flow Cytometry

A single cell suspension was prepared using Tumor Dissociation Kit and gentleMACS Octo Dissociator (Miltenyi Biotec, Bergisch Gladbach, Germany) and subjected to flow cytometry analysis to assess tumor infiltrating immune cells. Cells were pre-labelled with Zombie-NIR viability dye (BioLegend, San Diego, CA, USA) and stained where indicated with antibodies against FAS, PD-L1, CD80, CD86, intercellular adhesion molecule 1 (ICAM1), H-2K^d^/H-2D^d^, I-A/I-E, CD45, CD31, CD3, CD4, CD8a, CD11c, F4/80, CD107a, CD206, Gr1, forkhead box P3 (FoxP3), Granzyme B (GzmB) (all from BioLegend), CD11b (BD Biosciences, San Jose, CA, USA), or calreticulin (CRT, Novus Biologicals, Littleton, CO, USA). To assess the level of apoptosis, cells were stained with Annexin V apoptosis detection kit (BioLegend). Cells were analyzed using a LSRFortessa flow cytometer (BD Biosciences).

### 2.10. RT-qPCR

Total RNAs were extracted from the tumor cells using PureLink RNA Mini Kit (Ambion, Austin, TX, USA). Isolated RNA was reverse transcribed using Superscript IV kit (Life Technologies, Carlsbad, CA, USA). TaqMan assay was performed using TaqMan assay probes (Life Technologies, Carlsbad, CA, USA) in triplicates. Raw data was analyzed using StepOne Software v2.3 via ΔΔCt method and relative mRNA expression was normalized to *Gapdh*.

### 2.11. Detection of miR-22

Small RNAs were extracted from the tumor cells using mirVana miRNA isolation kit (Ambion, Austin, TX, USA). Isolated small RNA was reverse transcribed with TaqMan microRNA reverse transcription kit (Life Technologies, Carlsbad, CA, USA) using RT primers specific to either *miR-22* or *U6* snRNA (Life Technologies, Carlsbad, CA, USA). RT-PCR was performed using TaqMan small RNA assays for *miR-22* and *U6* snRNA (Life Technologies, Carlsbad, CA, USA). Raw data was analyzed using StepOne Software and relative expression of *miR-22* was normalized to the corresponding *U6* snRNA expression.

### 2.12. Bioluminescence Imaging

To measure lung metastatic LL/2-luc tumor growth, non-invasive bioluminescence imaging (BLI) was performed with the Xenogen IVIS in vivo imaging system (Caliper Life Sciences, Waltham, MA, USA). Mice were injected with XenoLight D-Luciferin (2.25 mg/mouse/IP, Perkin Elmer, Waltham, MA, USA) and were anesthetized during imaging using isoflurane (Piramal Healthcare, Aurora, ON, Canada). Bioluminescence intensity was quantified with Living Image software (Caliper Life Sciences, Waltham, MA, USA).

### 2.13. Transcriptome Analysis

Gene expression in the tumor samples was measured using commercially available gene panels (PanCancer Pathways and Immune Profiler, NanoString Technologies, Seattle, WA, USA). After normalization based on the geometric mean of negative controls, internal housekeeping genes, and positive controls, raw data was processed using nSolver 4.0 software (NanoString Technologies) and normalized counts were log2-transformed for further analysis. Genes that were increased or decreased by at least 50% from the baseline and false discovery rate (FDR)-adjusted *p* < 0.05 were considered differentially expressed genes. Gene ontology (GO) analyses were performed using the gene set enrichment analysis (GSEA) online tool (https://www.gsea-msigdb.org (accessed on 13 June 2022)).

### 2.14. Patient Survival Data Analysis

Publicly available lung cancer patient datasets (GSE31210, GSE30219 and GSE72094) were accessed and analyzed through Lung Cancer Explorer (LCE, https://lce.biohpc.swmed.edu/lungcancer (accessed on 26 May 2022)) to assess the association between overall survival (OS) and expression of *Gal-1* mRNA. Survival curves were estimated by LCE’s survival analysis module using the Kaplan–Meier method (survival, R package [14]). The survival analysis module of LCE also carried out a log-rank test and a Cox proportional hazard regression to assess the survival association and calculate the hazard ratio (HR) with *Gal-1* expression in each individual dataset [15].

### 2.15. Statistical Analysis

Statistical analyses were performed using SigmaPlot 11.2 (Systat Software, San Jose, CA, USA). Presented data represent mean ± standard error of the mean (SEM). Data normality was confirmed using the Shapiro–Wilk test. Student’s *t*-test or one-way analysis of variance (ANOVA) test followed by a Bonferroni-corrected Student’s *t*-test was carried out to compare data between two or more groups, respectively. Correlation analysis of *Gal-1* mRNA and *miR-22* level were carried out using Sigmaplot and data were tested for normality prior to the correlation analysis using the Shapiro–Wilk test. Survival analyses by the Kaplan–Meier method were compared by log-rank test. For all comparisons, *p* < 0.05 was considered significant.

## 3. Results

### 3.1. Combining Anti-PD-1 and SGT-53 Inhibits LL/2 Tumor Growth

To investigate if adding SGT-53 can overcome the resistance in anti-PD-1 therapy, C57BL/6 mice with subcutaneously established syngeneic LL/2 tumors were treated with either anti-PD-1 alone, SGT-53 alone, or combination of both agents (Figure 1A). No significant inhibition of tumor growth was seen with single-agent anti-PD-1 compared with untreated control (*p* > 0.05) suggesting that LL/2 tumors are primarily resistant to anti-PD-1 (Figure 1B). A treatment with SGT-53 alone resulted in only a modest growth inhibition. However, combining SGT-53 with anti-PD-1 significantly retarded tumor growth leading to considerably smaller tumors compared to those in mice given either agent as monotherapy (Figure 1B,C). In mice treated with SGT-53, either alone or in combination with anti-PD-1, an IHC analysis revealed a significant inhibition of tumor cell proliferation as indicated by the loss of Ki-67 expression compared to tumors in untreated mice or those treated with anti-PD-1 alone (Figure 1D,E). However, treatment with anti-PD-1 alone resulted in no changes in the level of Ki-67 expression. We also observed a significant increase of tumor cell apoptosis as indicated by expression of Casp3 in the tumors from mice receiving the combination treatment (Figure 1D,F). These data indicate that SGT-53 can convert otherwise unresponsive mouse NSCLC tumors into anti-PD-1-responsive tumors resulting in reduced tumor growth and increased tumor cell apoptosis.

### 3.2. Combining Anti-PD-1 and SGT-53 Improves Survival in a Metastatic LL/2 Tumor Model

We performed a survival study to assess the efficacy of combining anti-PD-1 and SGT-53 using a metastatic syngeneic NSCLC model in C57BL/6 mice. Six days post-injection of luciferase expressing LL/2-luc cells, the presence of lung metastatic tumors was confirmed by BLI. Starting on nine days after tumor injection, mice were given either anti-PD-1 alone, SGT-53 alone, or combination of both agents similar to above subcutaneous model, and tumor growth was monitored using BLI. As had been seen above in the subcutaneous model, an enhanced antitumor activity was evident with combining anti-PD-1 and SGT-53 (Figure 2A,B). BLI-based measurement of tumor size on days 18 and 25 revealed a significant inhibition of tumor growth associated with the combination treatment regimen (Figure 2B). However, anti-PD-1 alone showed no appreciable antitumor activity, and SGT-53 alone showed only a modest effect when compared with untreated animals. All untreated mice succumbed to their disease prior to day 26, and anti-PD-1 therapy provided no survival benefit (i.e., 24 days of median survival time compared to 23 days in untreated group, Figure 2C). However, the median survival of mice receiving SGT-53 plus anti-PD-1 treatments was significantly increased to 32 days. Collectively, these observations suggest that the combination of anti-PD-1 plus SGT-53 was more efficacious in conferring a survival benefit as an immunotherapy regimen than anti-PD-1 alone.

### 3.3. Combining Anti-PD-1 and SGT-53 Enhances Host Immune Responses

To investigate whether the SGT-53-mediated enhancement of tumor responses in the context of anti-PD-1 therapy are associated with enhanced host immunity, we assessed the expression of immune-related genes using NanoString nCounter gene expression assays. The transcriptomic changes in tumors receiving anti-PD-1 plus SGT-53 versus the baseline of single-agent anti-PD-1 therapy were illustrated in a volcano plot (Figure 3A). We have applied the criteria of at least 50% change compared to the baseline and FDR-adjusted *p* < 0.05 to define differentially expressed genes. As a result, a total of 28 genes (26 upregulated and two downregulated) were found to be differentially expressed between the tumors receiving anti-PD-1 alone or anti-PD-1 plus SGT-53 (Figure 3B). The genes that were upregulated in combination treatment group included genes related to interferon (IFN) responses (*Bst2*, *Cmpk3*, *Ifit3*, *Ifit4*, *Irf7*, *Oas2*, *Oasl1*, *Txk*, and *Zbp1*), the complement pathway (*ApoE*, *C3*, *C4b*, *Cfb*, and *Serping1*), immune cell activation and proliferation (*Fcgr4*, *Il2*, *Nfatc2*, and *Tnfrsf13b*), and antigen presentation (*H2-Aa*, *H2-Eb1*, and *Cd74*). Expression of *Cxcl1* and *Il23a* were downregulated in tumors from the combination treatment group compared to tumors from the anti-PD-1 group. To further understand the molecular or biological pathways that are impacted by the combination treatment regimen, we performed gene ontology (GO) analyses using the GSEA online tool to reveal several enriched pathways (Figure 3C). Enriched GO biological processes included regulation of immune system process, innate immune response, and adaptive immune response. Enriched GO cellular components included blood microparticle, major histocompatibility complex (MHC) protein complex, and endocytic vesicles. Enriched GO molecular functions included MHC protein binding, immune receptor activity, and cytokine activity. Enriched GSEA hallmarks included interferon responses, allograft rejection, complement, and inflammatory response. Enriched pathways included those involved in the interferon signaling, complement, and cytokine signaling in immune system (Figure 3C). Collectively, these data indicate that combining SGT-53 with anti-PD-1 appears to improve immune responses by affecting various pathways of antitumor immunity of the host.

To further probe the SGT-53-mediated enhancement of antitumor immunity, we quantified tumor-infiltrating immune cells including tumor-killing cytotoxic T lymphocytes (CTLs) and immunosuppressive regulatory T cells (Tregs) and myeloid derived suppressor cells (MDSCs). In animals receiving SGT-53, either alone or in combination with anti-PD-1, IHC analysis of subcutaneously established LL/2 tumors demonstrated increased infiltration of CD8 T cells compared to tumors in untreated mice or mice treated with anti-PD-1 alone (Figure 4A). In addition, compared with untreated tumors, tumors treated with SGT-53 showed increased mRNA expression of chemokines including *Cxcl9*, *Cxcl10*, and *Cxcl12* (Figure 4B), which are associated with increased infiltration of activated T cells in various human cancers via the chemokine receptors CXCR3 and CXCR4 [16]. It should be noted that CXCR3-dependent T cell infiltration is a prerequisite to the success of PD-1/PD-L1 blockade therapy [17]. Flow cytometry analyses revealed substantially increased granzyme B (GzmB)-positive CD8 T cells (CTLs) in tumors receiving SGT-53, either alone or in combination with anti-PD-1 compared to tumors in untreated mice or mice receiving anti-PD-1 alone (Figure 4C). Importantly, the number of FoxP3-positive CD4 T cells (Tregs) was significantly decreased by the combination of SGT-53 and anti-PD-1 compared with tumors given anit-PD-1 alone (Figure 4D). Similarly, immunosuppressive MDSCs were also significantly reduced by SGT-53 treatment, either alone or in combination with anti-PD-1, compared with tumors in untreated mice or mice treated with anti-PD-1 alone (Figure 4E). These data indicate that combining SGT-53 with anti-PD-1 can alter the TME to be less immunosuppressive and thereby boost the host’s antitumor immunity by increasing immune cell infiltration and activation.

Macrophages also play an important role in antitumor immunity [18]. Specifically, antitumoral M1 macrophages can inhibit tumor growth, while alternatively activated pro-tumoral M2 macrophages create an immunosuppressive TME to promote tumor growth. Accordingly, we have examined the effects of SGT-53 on tumor-associated macrophage phenotypes. RT-qPCR analysis of tumors revealed that SGT-53 treatment significantly downregulated the genes associated with M2 macrophage polarization (*Arg1*, *Snail*, and *Cox2*), while increasing the expression of M1-associated *Nos2* (Figure 4F). Flow cytometry analyses further confirmed the significant decrease of M2 macrophages in LL/2 tumors after SGT-53 treatment when compared to tumors from untreated animals (Figure 4H), while M1 macrophages remained largely unchanged across all the treatment groups (Figure 4G). As a result, SGT-53 significantly increased the M1/M2 ratio, while anti-PD-1 treatment did not have a significant effect on this ratio (Figure 4I). Thus, our results indicate that SGT-53 treatment alters the balance of tumor-associated macrophage phenotypes toward lowering immunosuppression, and this effect may contribute to the inhibition of tumor progression by SGT-53 when given in the context of anti-PD-1 immunotherapy.

### 3.4. SGT-53 Increases Immunogenicity of Tumor Cells

To assess whether overexpression of p53 altered LL/2 cell survival via enhanced apoptosis, a known canonical function of p53, we performed Annexin V assay (Figure 5A). Both Annexin V^+^/7AAD^−^ (apoptotic) and Annexin V^+^/7AAD^+^ (dead) cells were significantly increased at 48 h after SGT-53 treatment compared to untreated cells. In contrast, transfection with the control nanocomplex loaded with a plasmid encoding the green fluorescent protein (*GFP*, scL-GFP) did not result in significant cell death demonstrating that the response to SGT-53 is not a result of nonspecific DNA-mediated cytotoxicity. To assess if SGT-53 treatment could induce immunogenic cell death (ICD), we examined three hallmarks of ICD in vitro. Flow cytometry analysis of LL/2 cells revealed increased surface expression of calreticulin (ecto-CRT) 48 h after SGT-53 treatment (Figure 5B). Analysis of cell culture media showed increased release of ATP and HMGB1 24 h after SGT-53 treatment, while control nanocomplex loaded with the control plasmid vector without *p53* (scL-vec) did not increase any of these markers (Figure 5C,D). Together, these data indicate that expression of functional p53 is responsible for induction of ICD to increase the immunogenicity of LL/2 cells.

We also investigated the impact of SGT-53 treatment to alter immunogenicity of tumor cells in vivo. Flow cytometry analyses of LL/2 tumors revealed a significantly increased surface expression of immune cell recognition molecules including FAS, MHC class I, PD-L1, ICAM1, CD80 and CD86 emanating from SGT-53 treatment (Figure 5E). RT-qPCR assay further revealed that SGT-53 treatment was able to increase mRNA levels of several components of the antigen-presentation machinery (*MHC I*, *Tap1* and *Tap2*) and *Pd-l1*, while downregulating genes linked to immunosuppression (*Ido1* and *Tgfb1*) (Figure 5F). Our data indicate that expression of functional p53 is responsible for both induction of ICD and enhancement of the immunogenicity of LL/2 cells, and these changes would be expected to increase the impact of anti-PD-1.

### 3.5. SGT-53 Represses Immunosuppressive Gal-1

To further understand mechanisms underlying our observations related to SGT-53 sensitizing LL/2 tumor to anti-PD-1, we have studied Gal-1, a glycan-binding protein with known immunosuppressive functions [19]. Notably, tumor-derived Gal-1 has been shown to induce T cell apoptosis and immune escape as well as tumor progression and metastasis [20,21]. Importantly, survival analysis of publicly available microarray data of lung cancer patients revealed that the high level transcriptional expression of *Gal-1* is associated with the poor OS for lung cancer patients in GSE31210, GSE30219, and GSE72094 datasets (Figure 6A). To understand the consequence of p53 induction on *Gal-1* expression, LL/2 cells were transfected with SGT-53 and expression of *Gal-1* was assessed in vitro. When p53 expression in tumor cells was enhanced by SGT-53, we observed a significant decrease of *Gal-1* at both the mRNA and protein levels while a control transfection with scL-vec nanocomplex did not change *Gal-1* levels (Figure 6B,C). A similar repression of *Gal-1* expression was observed in human NSCLC cell line H358 after SGT-53 treatment (Figure 6D). To validate the role of *Gal-1* causing a therapeutic resistance to anti-PD-1 therapy, siRNA-mediated knockdown of *Gal-1* using our scL nanocomplex was tested in combination with anti-PD-1 therapy in a metastatic LL/2-luc tumor model in C57BL/6 mice (Appendix A). Strikingly, similar to anti-PD-1 and SGT-53 combination study, BLI-based measurement revealed a significant inhibition of tumor growth associated with the anti-PD-1 and scL-siGAL combination treatment (Appendix A) with a significant survival benefit (Appendix A). Collectively, these observations suggest that the *Gal-1* plays a critical role in the therapeutic resistance to anti-PD-1 therapy and suppression of *Gal-1* by either siRNA or SGT-53 could reverse the resistance.

Interestingly, SGT-53 treatment significantly increased *miR-22* (Figure 6E), which has been shown to inhibit the expression of *Gal-1* in hepatocellular carcinoma and renal cell carcinoma [22,23]. When we correlated levels of *Gal-1* mRNA (Figure 6B) and *miR-22* (Figure 6E), we observed an inverse correlation between them (Figure 6F) suggesting that *Gal-1* downregulation might be mediated through *miR-22* upregulation by p53. IHC analysis of LL/2 tumors from mice receiving SGT-53 treatment, we saw a significant decrease of Gal-1 positive staining compared with tumors from untreated mice (Figure 6G). When compared to plasma from healthy non-tumor bearing mice, plasma from mice bearing LL/2 tumor exhibited significantly higher levels of Gal-1 protein, and SGT-53 treatment was able to lower it substantially to the level resembling those in healthy non-tumor bearing mice (Figure 6H). Since hypoxia-inducible factor 1-alpha (*HiF1a)* has also been reported to upregulate *Gal-1* expression [24], we have also examined *Hif1a* mRNA expression in LL/2 tumors. RT-qPCR analysis of tumors confirmed that SGT-53 treatment can significantly downregulate expression of both *Gal-1* and *Hif1a* mRNA (Figure 6I). Taken together, our results indicate that p53 overexpression can downregulate immunosuppressive *Gal-1* via *miR-22* upregulation and/or *Hif1a* downregulation. This inhibition of *Gal-1* might be, in part, responsible for the observed reversal of anti-PD-1-resistance when SGT-53 is administered concurrently with anti-PD-1.

## 4. Discussion

The emergence of ICIs has shifted the paradigm of cancer treatment. Specifically, monoclonal antibodies targeting CTLA-4 and PD-1/PD-L1 have successfully elicited antitumor responses, and the FDA has approved several such antibodies for clinical use. However, the vast majority of patients do not experience benefit from ICIs, prompting numerous studies combining ICIs with various treatment modalities to improve efficacy of ICIs [7].

Designing rational combinations that provide maximal benefit is challenging because killing of tumor cells is regulated by many factors that together shape antitumor immune responses [25]. In recent years, one of the most important insights into tumor immunity was provided by the identification of immunosuppressive negative regulatory pathways and immune escape mechanisms that prevent the host immune responses from eradicating cancer cells [20,21]. Moreover, to oppose a CTL response that could eliminate tumors, cancer cells exploit various immune evasion mechanisms simultaneously. Consequently, it is unlikely that targeting a single resistance mechanism will prove adequate in eradicating tumors that are otherwise refractory to immunotherapy [25]. Thus, research continues aimed at elucidating strategies to overcome these immune evasion mechanisms and to restore the patient’s ability to recognize, target, and destroy tumor cells [26,27].

Here, we have combined *p53* gene therapy via tumor-targeted nanomedicine SGT-53 with anti-PD-1 therapy. SGT-53 is based on a nanotechnology platform (termed scL for single chain Liposome) for systemic delivery of anticancer therapeutics selectively to tumor cells. This scL nanoparticle is a self-assembled, biodegradable, immunoliposome that employs TfRscFv as a targeting ligand to take advantage of the elevated TfR expression found on most tumor cells [28]. Binding of scL nanocomplex to TfRs on tumor cells facilitates intracellular delivery of the payload via receptor-mediated endocytosis and the rapid recycling of TfR can increase the accumulation of payloads in tumor cells [13]. Moreover, systemic administration of SGT-53 nanocomplex as an anticancer nanomedicine, has been translated into completed Phase I clinical trials demonstrating very low toxicity with indications of anticancer effect in some patients [29,30]. SGT-53 is currently in a Phase II trial for advanced pancreatic cancer with some very promising interim results [31].

Accumulating evidence has demonstrated that tumor suppressor p53 broadly participates in modulation of antitumor immune responses [10]. A linkage between p53 and immune system was illustrated in very early studies using p53-null mice with nearly 25% of them dying from unresolved infections prior to tumor development indicating a defective immune system [32]. Moreover, loss of p53 function in cancers results in profound changes in chemokine/cytokine secretion, leading to a significant transformation of the immune landscape [33]. Cancer-specific deletion of p53 promoted the recruitment and instruction of suppressive myeloid cells and attenuated CTL responses by increasing suppressive Tregs and impairing GzmB in CD8+ T cells [11]. Therefore, manipulation of p53 pathway has a very strong potential to alter the immunological landscapes of tumors in a way that could be therapeutically beneficial. Indeed, we have demonstrated that restoration of p53 function via SGT-53 can improve anti-PD-1 immunotherapy of several immune-resistant mouse tumors including head and neck cancer, breast cancer, and glioma [34,35,36]. In this study, we have demonstrated the ability of SGT-53 to restore cytotoxic immune interactions and to enhance anti-PD-1 therapy using p53-mutant and ICI-resistant syngeneic LL/2 tumors and explored the molecular basis for these observations. Extrapolating our observations to human cancers, a conversion of immune-resistant NSCLC into tumors responsive to ICIs would have the potential to convert patients who do not respond to anti-PD-1 therapy into responders, thereby making checkpoint blockade clinically beneficial for a larger proportion of advanced NCSLC patients.

We have shown that restoration of p53 function via SGT-53 treatment changed immune responses in a multifaceted way as might be expected given that p53 is a pleiotropic regulator of many downstream genes. Firstly, SGT-53 treatment was able to induce the ICD of tumor cells, which is a crucial step in the initiation of T cell-mediated antitumor immunity [25]. Secondly, SGT-53 treatment increased immunogenicity of tumor cells including elevated expression of immune recognition molecules as well as antigen processing pathways, which are important in immune surveillance and recognition. In cancer cells, genetic perturbations within the antigen presentation machinery reduces immunogenicity and promotes immune escape [37]. Consistent with our observation, p53 increased antigen processing and surface MHC-I complexes [38]. Thirdly, SGT-53 treatment relieved immunosuppression while increasing CTL activity. SGT-53 treatment significantly increased expression of chemokines (*Cxcl9*, *Cxcl10*, and *Cxcl12*) directly associated with T cell migration and infiltration through chemokine receptors CXCR3 and CXCR4 [16]. In addition to the increased recruitment signals of immune effector cells, we have also observed that SGT-53 treatment resulted in a shift in macrophage polarization away from the immunosuppressive M2 phenotype and toward tumor-inhibiting M1 phenotype. It has been shown that myeloid cells in p53-null tumors are recruited and reprogrammed towards M2 macrophages [11]. In line with our observation, p53 activation by nutlin-3a treatment also decreased M2 polarization [39]. Besides M2 macrophages, Tregs are crucial in enforcing immunosuppression and inhibiting CTL responses. Of note, p53 regulates transcription of *FoxP3*, the transcription factor responsible for Treg differentiation to prevent autoimmunity [40]. However, a recent study demonstrated that Tregs are enriched in p53-null tumors, a response that is in part mediated through de novo generation of Tregs by PMNs [41], supporting a Treg inhibitory function of p53. Here, the combination of SGT-53 and anti-PD-1 significantly lowered Tregs suggesting negative regulation of Treg by p53, a function of p53 that warrants further investigation. It is possible that SGT-53 treatment downregulates tumor expression of *Tgfb*, which is a key regulator of the signaling pathways that initiate and maintain *FoxP3* expression and the suppressive function in CD4^+^CD25^−^ precursors [42]. Collectively, our results indicate that LL/2 tumor-bearing mice treated with SGT-53 have tumors that are more immunologically ‘hot’ with the multiple changes induced by SGT-53 likely combining to enhance immune responses during ICI therapy.

It is important to note that the tumor sensitization to anti-PD-1 therapy via SGT-53 treatment was accompanied by the repression of *Gal-1*. Interestingly, *Gal-1* along with *Tgfb* has been shown to play a critical role in conferring feto-maternal immune tolerance during pregnancy [43]. *Gal-1* is also involved in T cell selection in the thymus by directly inducing apoptosis of thymocyte subsets [44]. Importantly, Gal-1 is overexpressed and secreted into the surrounding milieu by many tumor types including NSCLC [12], and tumor-derived Gal-1 contributes toward immunosuppression and tumor-immune escape by causing apoptosis or exclusion of T cells [20,21]. Together, these observations highlight the relevance of Gal-1 as an emerging therapeutic target in lung cancer with potential to block immunosuppression and tumor progression. In fact, it has been shown that knockdown of *Gal-1* using shRNA in mouse lung cancer cells decreased the number of metastatic tumor nodules in the lungs [45]. DNA aptamer targeting *Gal-1* rescued T cells from apoptosis and restored T cell-mediated immunity in syngeneic LL/2 tumor [19]. In our study, we report that systemic administration of SGT-53 appears to negatively regulate the level of Gal-1 expression in tumor. Consistent with our observation, downregulation of Gal-1 was shown in human glioma cells after treatment with adenoviral vector carrying exogenous wild-type p53 [46]. These results indicate that restoration of functional p53 could reverse the immunosuppression mediated by Gal-1 and stimulate CTL response to increase anti-PD-1 efficacy. Taken together, all of these data support the idea that inhibition of Gal-1 is, in part, responsible for the observed reversal of anti-PD-1-resistance by SGT-53 treatment.

Although we have shown that SGT-53 treatment can inhibit Gal-1 in NSCLCs cells, the mechanism of this inhibition requires further exploration. Currently, convincing evidence of a direct functional association between Gal-1 and p53 are scant [47]. One possible explanation is that p53 increases expression of *miR-22* that, in turn, downregulates Gal-1. This particular microRNA is a direct transcriptional target of p53 and wild-type p53 binds to the promoter region of *miR-22* to enhance its expression [48]. In line with our data, as an upstream regulator of Gal-1 [49], *miR-22* was shown to inhibit Gal-1 in hepatocellular carcinoma [22] and renal cell carcinoma where transfection with *miR-22* mimics resulted in a dose-dependent decreases of Gal-1 expression [23]. Furthermore, a sequestration of *miR-22* by circular RNA fibroblast growth factor receptor 3 (*FGFR3)* increased Gal-1 level and consequently promoted NSCLC progression [50]. Another route of SGT-53 modulating Gal-1 could be via HIF1a. In agreement with our data, it has been shown that p53 can inhibit HIF1 activity by targeting the HIF1a subunit for Mdm2-mediated ubiquitination and proteasomal degradation, and that loss of p53 enhances hypoxia-induced HIF1a levels in tumor cells [51]. Furthermore, recent studies suggested that HIF1a protein can increase Gal-1 expression in colon cancer by binding to DNA to promote transcription of the *Gal-1* gene, whereas shRNA-mediated silencing of HIF1a expression antagonized hypoxia-induced Gal-1 expression [24]. Interestingly, *miR-22* was shown to modulate HIF1a expression in human colon cancer cells where overexpression of *miR-22* repressed HIF1a expression and knockdown of endogenous *miR-22* enhanced HIF1a expression conversely [52]. Taken together, it is possible that p53 could modulate Gal-1 expression through tumor suppressive *miR-22* and/or HIF1a [23].

## 5. Conclusions

In conclusion, we herein provide evidence showing that SGT-53 contributes to the sensitization of mouse NSCLCs to anti-PD-1 therapy by reducing immunosuppression and restoring antitumor immune responses. Our findings provide a compelling rationale for evaluation of the combination of p53 gene therapy and anti-PD-1 immunotherapy in patients with advanced NSCLC that are otherwise resistant to immunotherapy.

## Figures and Tables

**Figure 1 cells-11-03434-f001:**
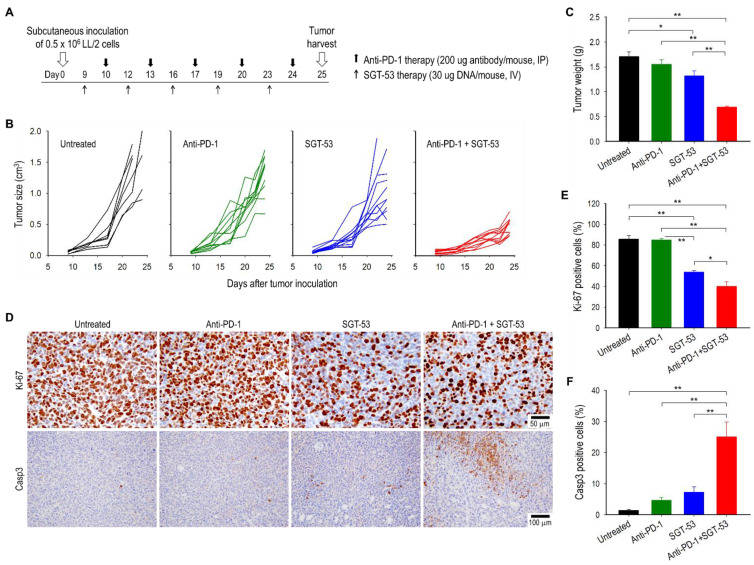
Combination treatment of anti-PD-1 and SGT-53 inhibits NSCLC growth. C57BL/6 mice with subcutaneous LL/2 tumors were randomized to therapy with anti-PD-1 (200 µg, IP) only, SGT-53 (30 µg DNA, IV) only, or the combination of both agents. (**A**) Treatment schedule. (**B**) Changes in individual tumor sizes. (**C**) Tumor weight at harvest on day 25. (**D**) IHC examination of tumors. Representative staining of Ki-67 and Casp3 are shown. Quantifications of positive stain of (**E**) Ki-67 and (**F**) Casp3. Data are presented as mean ± SEM and were analyzed by one-way ANOVA. * *p* < 0.05, ** *p* < 0.001. *n* = 7–10/group.

**Figure 2 cells-11-03434-f002:**
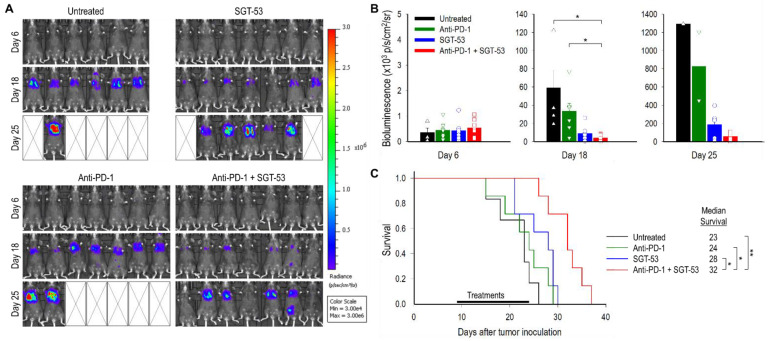
SGT-53 enhances the efficacy of anti-PD-1 therapy in a metastatic LL/2-luc tumor model. (**A**) Bioluminescence images (BLI) of metastatic lung cancer are shown. Bioluminescence signals, shown in a color map, correlate with tumor sizes. Red color: stronger signal, violet color: weaker signal. (**B**) Bioluminescence intensities of tumors were plotted. Circles, squares, and triangles indicate individual mouse. Data are presented as mean ± SEM and were analyzed by Student’s *t* test. * *p* < 0.05. (**C**) Kaplan–Meier survival curves of mice. Log-Rank test, * *p* < 0.05, ** *p* < 0.001. *n* = 6–7/group.

**Figure 3 cells-11-03434-f003:**
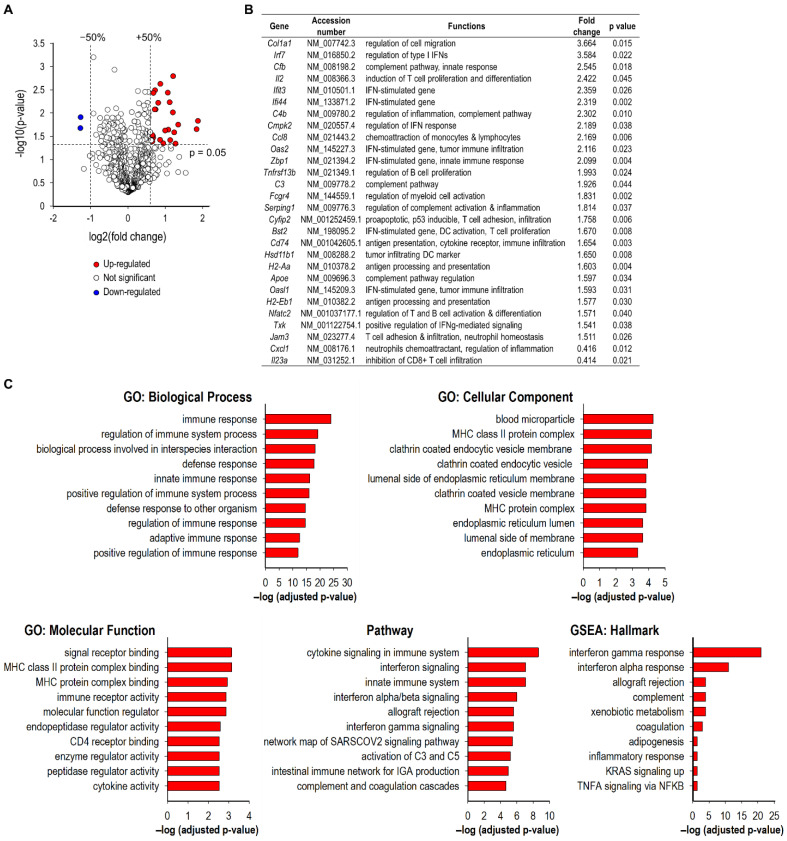
Transcriptomic analyses of tumors using NanoString Immune Profiling assay. (**A**) Volcano plot showing differential expression of immune related genes in anti-PD-1 plus SGT-53 versus the baseline of anti-PD-1 monotherapy. Volcano plot is presented as fold change in gene expression [log2(fold change)] against significance of change [−log10(*p*-value)]. (**B**) The list of genes with a significant change. (**C**) GSEA analysis of 28 identified differentially expressed genes. Groups reflect main categories of GO terms: biological process; cellular component; molecular function, hallmark, and canonical pathway. Vertical and horizontal axes represent each category term and −log10(FDR adjusted *p*-value) of the corresponding term, respectively.

**Figure 4 cells-11-03434-f004:**
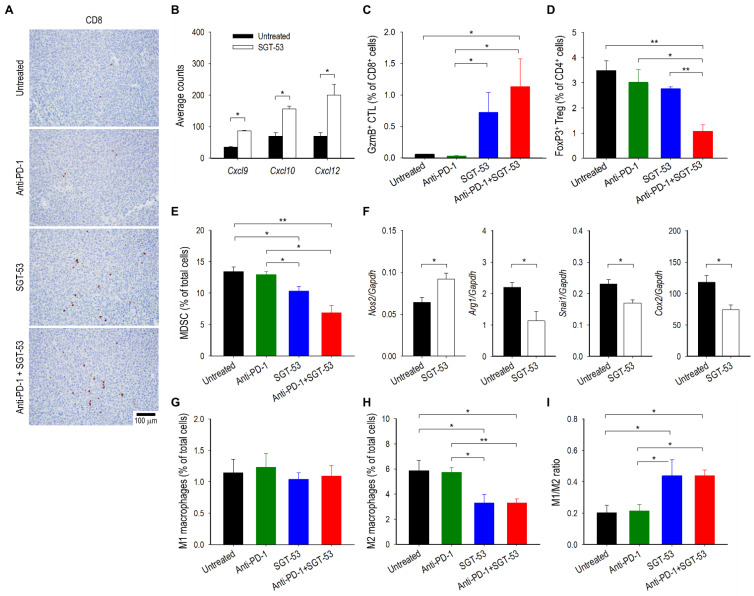
Treatment of anti-PD-1 and SGT-53 enhances host immune responses. Mice with LL/2 tumors were treated with anti-PD-1 (200 µg), SGT-53 (30 µg DNA), or the combination of both agents. (**A**) IHC examination of tumors. Representative staining of CD8 is shown. (**B**) Changes in expression of chemokines related to T cell migration were assessed using NanoString analysis. Tumor-infiltrating immune cells including (**C**) CTLs (CD45^+^CD3^+^CD8^+^GzmB^+^), (**D**) Tregs (CD45^+^CD3^+^CD4^+^FoxP3^+^) and (**E**) MDSCs (CD45^+^CD11b^+^Gr1^+^) were assessed via flow cytometry. (**F**) Expressions of genes associated with macrophage polarization in the whole tumor were assessed by RT-qPCR. Quantifications of populations of (**G**) M1 macrophage (CD45^+^CD11b^+^F4/80^+^CD11c^+^) and (**H**) M2 macrophage (CD45^+^CD11b^+^F4/80^+^CD206^+^) in tumor using flow cytometry. (**I**) The ratio of M1/M2 macrophages. Data are presented as mean ± SEM and were analyzed by one-way ANOVA or Student’s *t* test. * *p* < 0.05, ** *p* < 0.001. *n* = 5–10/group. Flow cytometry gating strategy and representation are provided in Appendix A.

**Figure 5 cells-11-03434-f005:**
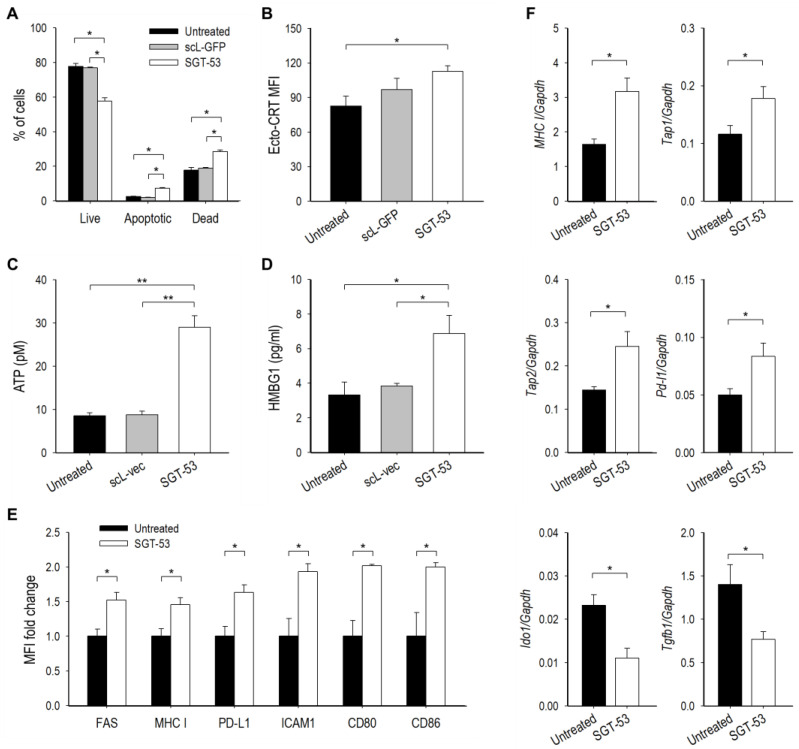
SGT-53 increases immunogenicity of tumor cells both in vitro and in vivo. For in vitro study, LL/2 cells were transfected with SGT-53 or a tumor-targeting nanocomplex loaded with either a plasmid encoding *GFP* (scL-GFP) or an empty plasmid vector (scL-vec). (**A**) Induction of apoptosis was assessed via Annexin V/7-AAD staining. *n* = 5/group. (**B**) Surface expression of ecto-CRT was assessed via flow cytometry. The mean fluorescence intensity (MFI) was plotted. *n* = 3/group. Release of (**C**) ATP and (**D**) HMGB1 was assessed in culture media. *n* = 3–9/group. For in vivo study, mice with subcutaneous LL/2 tumor received SGT-53 treatment. (**E**) Flow cytometry analysis of cell surface components of immunogenicity. Tumor cells were identified using CD45^−^ live cell gate. The MFI fold changes were plotted in comparison with those in tumors from untreated mice. *n* = 3/group. (**F**) Expression of genes associated with antigen presentation and immunosuppression was assessed by RT-qPCR. *n* = 5–10/group. Data are presented as mean ± SEM and were analyzed by Student’s *t* test. * *p* < 0.05, ** *p* < 0.001. Flow cytometry gating strategy and representation are provided in Appendix A.

**Figure 6 cells-11-03434-f006:**
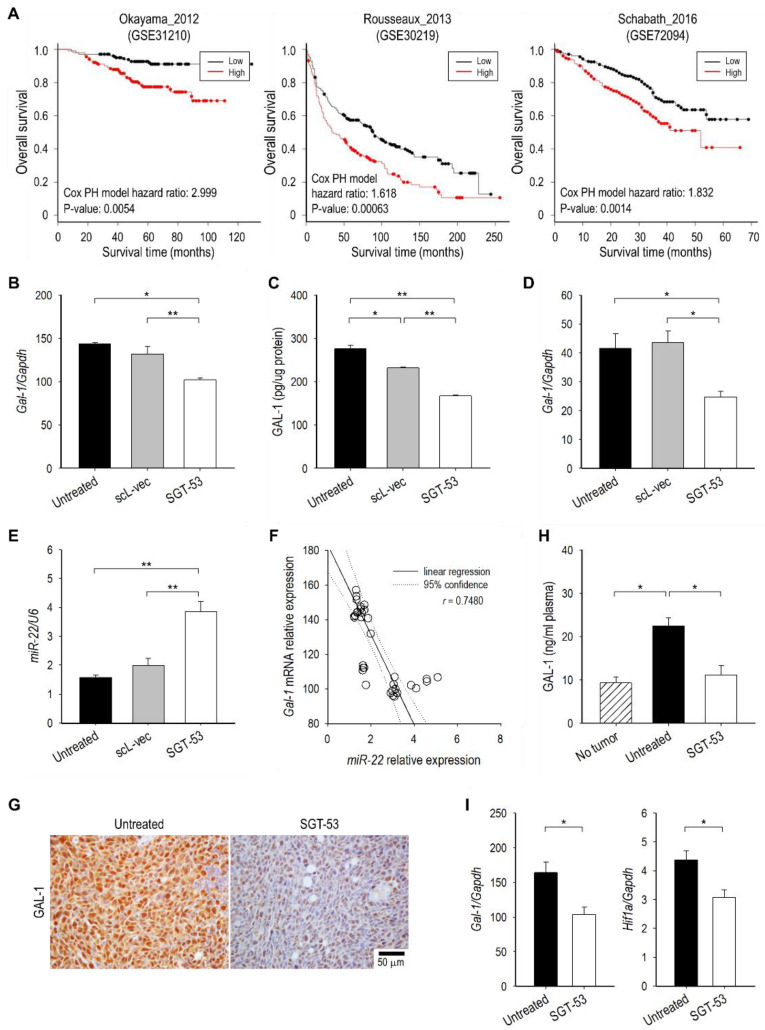
SGT-53 treatment represses immunosuppressive Gal-1. (**A**) Survival analysis of Gal-1 expression in lung cancer patients. Gal-1 expression in LL/2 cells transfected with either SGT-53 or scL-vec was assessed at (**B**) the mRNA level by RT-qPCR and (**C**) the protein level by ELISA immunoassay. (**D**) Expression of *Gal-1* was assessed in human NSCLC cell H358 by RT-qPCR. (**E**) Quantitative analysis of *miR-22* in LL/2 cells using RT-qPCR. (**F**) The correlation between *miR-22* and *Gal-1* mRNA levels in LL/2 cells. The circles indicate individual sample. (**G**) IHC examination of Gal-1 expression in LL/2 tumors. Representative images of Gal-1 staining are shown. (**H**) Quantitative analysis of Gal-1 protein in plasma using ELISA immunoassay. (**I**) Expression of *Gal-1* and *Hif1a* was assessed in LL/2 tumors by RT-qPCR. Data are presented as mean ± SEM and were analyzed by Student’s *t* test. * *p* < 0.05, ** *p* < 0.001.

## Data Availability

Not applicable.

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
