# Peer review of "A Novel P53 Nanomedicine Reduces Immunosuppression and Augments Anti-PD-1 Therapy for Non-Small Cell Lung Cancer in Syngeneic Mouse Models"

_cells, 2022, doi:10.3390/cells11213434_

Round 1

Reviewer 1 Report

The current paper entitled “A Novel P53 Nanomedicine Reduces Immunosuppression and 2 Augments Anti-PD-1 Therapy for Non-Small Cell Lung Cancer” indicated the significance of p53 gene therapy via a tumor-targeting nanomedicine (termed SGT-53) on the efficacy of anti-PD-1 therapy in lung cancer. The manuscript is very informative and is well written though minor concerns exist.

1-    Based on section 2.9, I assumed that cell separation was conducted; however, I cannot find related section for this part in results.

2-    Can author explain whether RT-PCR analysis of genes associated with M2 macrophage polarization obtained from whole tumor or isolated macrophages.

Reviewer 2 Report

1.    Several manuscripts are available where people showed how the combination of SGT-53 and the anti-PD1 antibody suppresses tumor growth in breast cancer, non-small cell lung carcinoma, glioblastoma, etc. In this manuscript, author used the same type of non-small cell lung carcinoma cells i.e., LL/2, which was already done by other groups. So, in my opinion, there is no novelty in this paper. The only thing which is very interesting and new in this paper is, How SGT-53 suppressed the function of immunosuppressive glycan binding protein Gal-1. But author didn’t show enough data which will validate this finding. Author showed focused more on the pathway related to Gal-1 and how SGT-53 along with the anti-PD-1 antibody suppresses it,  instead of validating the tumor suppressive function of SGT-53 and anti-PD-1 antibody, which is already well known.

2.    For Figure 1. Author should provide the mouse tumor images of the subcutaneously injected LL/2 cells.

3.    For figure 2. As LL/2 cells produce lung Mets, it’s better to show lung Mets with MFI and Lung histology.

4.    In Figure 2, mouse numbers are not consistent. Why? In the untreated group, almost 90% of animals died on day 25. Author should reconsider the time points for this experiment as making any conclusion is very hard due to the higher mortality of the control group. Author mentioned SGT-53 enhances the efficiency of anti-PD1, but in the only anti-PD1 group, more than 50% of animals died. Author should explain this.  

5.    For figure 4, author should provide flow representation (along with full gate) for 4C, 4D, 4E, 4D and 4H.

6.    In Figure 5, should add flow gates with representation. Why author suddenly removed the anti-PD1 group?

7.    In figure 6, author transfected SGT-53 in the lung tumor cells and showed decreased expression of Gal-1 expression in vitro. Is the metastasis effect of those tumor cells also affected? Author should add more experiments like migration, etc.  

8.     Author should use those SGT-53 transfected cells in the mouse model to see how those cells migrate and form tumor in the lungs.  

9.    For figure 6Its good to have an alone anti-PD1 group with transfected SGT-53 cells in vivo.

10. Author should provide more data related to Gal-1 experiment.

Reviewer 3 Report

Thank you for inviting me to review the manuscript “A Novel P53 Nanomedicine Reduces Immunosuppression and Augments Anti-PD-1 Therapy for Non-Small Cell Lung Cancer”. I found this topic interesting and worth exploring, considering the unsatisfactory response to anti-PD-1 therapy still observed in clinical settings. In the manuscript, the Authors presented the utility of p53 gene therapy in increasing the efficacy of PD-1 inhibitors. Overall, the manuscript is very well written, the study was well-planned, and the methods and results were presented in detail. The Authors put a lot of effort into conducting multiple in vitro and in vivo experiments to prove their hypothesis, and obtained interesting and promising results. Below, please find several comments that could improve the manuscript.

Major issues:

(1)   Line 1: I would suggest modifying the manuscript title a little with the indication that the experiments were conducted using animal models; right now, it is unclear whether the study regards human subjects, animals, or only in vitro experiments.

(2)   Lines 75-90: The paragraph should be shortened; the Authors summarized all received results, which would better fit the ‘Conclusions’ section.

(3)   Line 111: Information about plasma/tumor tissue collection is missing in the ‘Animal Studies’ section. How were the samples handled (stored) after collection? When were the plasma samples collected? Was it EDTA or heparin plasma?

(4)   Line 183: The Authors used ANOVA and Student’s T-test for data comparison. Was the normal distribution of variables confirmed (and what test was used)? Also, based on the presented data, a few more tests were used that were not mentioned in the ‘Statistical Analysis’ section (like Cox regression analysis or correlation tests); please revise this section.

(5)   Line 194: How was no inhibition/inhibition assessed (Figure 1B)? Visually or by statistical tests? Please, clarify.

Minor issues:

(1)   Genes should always be italicized; please, correct (e.g., lines 37, 39, 253, …).

(2)   Please, introduce abbreviations for EGFR, IP, and IV.

(3)   Line 42: NSCLC cannot be “an unmet medical need”; its successful/effective treatment can. Please, rephrase the sentence.

(4)   Lines 45-47: I would also add PD-L1 inhibitors as they belong to ICIs.

(5)   IP/i.p., IV/i.v., and SE/SEM abbreviations were used interchangeably throughout the manuscript; please, unify.

Round 2

Reviewer 2 Report

Author addressed all the concerns. I am recomanding this paper for publication.